# Molecular Mechanisms Underlying Initiation and Activation of Autophagy

**DOI:** 10.3390/biom14121517

**Published:** 2024-11-27

**Authors:** Zhixiao Wei, Xiao Hu, Yumeng Wu, Liming Zhou, Manhan Zhao, Qiong Lin

**Affiliations:** School of Medicine, Jiangsu University, 301 Xuefu Road, Zhenjiang 212013, China; 2212213070@stmail.ujs.edu.cn (Z.W.); 2212213071@stmail.ujs.edu.cn (X.H.); 2212213095@stmail.ujs.edu.cn (Y.W.); 2212313098@stmail.ujs.edu.cn (L.Z.); 2212313096@stmail.ujs.edu.cn (M.Z.)

**Keywords:** autophagy initiation, liquid–liquid phase separation (LLPS), signaling pathways, atg9a

## Abstract

Autophagy is an important catabolic process to maintain cellular homeostasis and antagonize cellular stresses. The initiation and activation are two of the most important aspects of the autophagic process. This review focuses on mechanisms underlying autophagy initiation and activation and signaling pathways regulating the activation of autophagy found in recent years. These findings include autophagy initiation by liquid–liquid phase separation (LLPS), autophagy initiation in the endoplasmic reticulum (ER) and Golgi apparatus, and the signaling pathways mediated by the ULK1 complex, the mTOR complex, the AMPK complex, and the PI3KC3 complex. Through the review, we attempt to present current research progress in autophagy regulation and forward our understanding of the regulatory mechanisms and signaling pathways of autophagy initiation and activation.

## 1. Overview of Autophagy

Autophagy is a highly conserved cellular catabolic process [1]. Since Yosuke Ohsumi discovered the key autophagic genes in yeast [2,3], numerous studies have uncovered the molecular events of autophagy from yeast to mammalian cells. Autophagy is divided into three types: macroautophagy, microautophagy, and chaperone-mediated autophagy (CMA) [4,5]. In microautophagy, cargo proteins are degraded through the direct invagination of lysosomal membranes [6], while in CMA, cargos bind to chaperones that mediate the entry of cargos into lysosomes for degradation [7]. In macroautophagy, cargos are wrapped by double-layer membrane vesicles (autophagosomes) and transported to the lysosome for degradation [8,9] (Figure 1). Autophagy, when mentioned in this article, refers to macroautophagy.

Macroautophagy or autophagy is classified into two types, non-selective and selective autophagy, based on the autophagic cargo types. In non-selective autophagy, the cargo, usually a portion of cytoplasm, is randomly wrapped into autophagosomes for lysosomal degradation to provide nutrition for cells. In selective autophagy, the autophagic cargo such as damaged organelles, misfolded proteins, invaded bacteria, or viral particles is recognized by autophagic receptors, recruited to autophagosomes, and transported to lysosomes for degradation [10,11]. As shown in Figure 1, the autophagic process is organized, driven, and regulated by a variety of autophagic signaling pathways and autophagy-related (ATG) proteins [12,13,14], including ULK1 complex (ULK1, FIP200, ATG13, ATG101), PI3KC3 complex (VPS34, VPS15, BECN1, ATG14L), ATG9A vesicle transport system (ATG9A and ATG2A), and two ubiquitin-like conjugation and ligation complexes: the ATG3-ATG7 complex and the ATG12-ATG5-ATG16L complex for LC3 conjugation and ligation with phosphatidylethanolamine (PE) [15,16,17]. The lipidation of ATG8 (or LC3 in mammalian cells) family proteins with PE was thought to be a specific process in autophagy for autophagosomal biogenesis. However, recent studies have found that the lipidation of the ATG8 family proteins occurs not only on autophagosomes, but also on other types of membranes, such as endosomes and lysosomes [18,19,20,21,22], and participates in other cellular processes, such as extracellular vesicle secretion and the repair of lysosomal membrane damage, in addition to autophagy [20,21]. ATG8 family proteins can also be lipidated with phosphatidylserine (PS) to conjugate to single membranes in non-canonical autophagy [23]. Furthermore, it was observed that ATG8 family proteins are conjugated with ATG3 by covalently linking to the lysine 243 residue of ATG3, similar to the reaction in protein ubiquitination [18], suggesting that ATG8 family proteins may function as a ubiquitin-like protein to directly modify other proteins. Thus, it is proposed that ATG8 family proteins, akin to ubiquitin, are a class of general membrane and protein modification molecules. This membrane or protein modification by ATG8 family proteins is designated as ATG8ylation or LC3ylation in mammalian cells [18,19]. ATG8ylation emerges as a new biomolecular modification mode and may play important roles not only in autophagy but also in many other cellular processes.

Autophagy is activated by the ULK1 complex. When the upstream signaling of autophagy, such as inhibition of mTOR or activation of AMPK, is elicited, the ULK1 complex is subsequently assembled, thereby activating the class III PI3K complex. The activated PI3K complex generates phosphatidylinositol 3-phosphate (PI3P) to recruit and assemble a pre-autophagosomal structure (PAS). At the same time, ATG9A provides membrane sources to the PAS for the formation of autophagosomes [24]. Apparently, signaling that activates autophagy is the key factor determining where and when the autophagy occurs. Therefore, the investigation of the mechanisms underlying the initiation and activation of autophagy is crucial for understanding the regulation and biological function of autophagy and targeting autophagy for the therapy of autophagy-associated diseases. This review will go through the recent findings about the initiation and activation of autophagy and hopefully provide insight into the mechanisms underlying the initiation and activation of autophagy.

## 2. Role of Liquid–Liquid Phase Separation (LLPS) in the Assembly of Pre-Autophagosomal Structure (PAS) During Initiation of Autophagy

Liquid–liquid phase separation (LLPS) refers to the process of forming phase-separated droplets with different components and properties through the interaction of certain biological macromolecules (such as proteins and RNA) in cells [25,26,27,28]. Current studies have found that proteins that drive LLPS usually have one or more intrinsically disordered regions (IDRs) [29], and the promotion of phase separation by IDRs may be related to their lack of hydrophobic amino acids [30]. The abnormality of LLPS is associated with a variety of diseases [31,32,33,34,35].

In recent years, more and more evidence shows that LLPS plays an important role in autophagy [36,37], especially in autophagy initiation and the formation of autophagosome precursors [38,39,40]. The mechanism underlying the LLPS-regulated assembly of pre-autophagosomal structures (PASs) in yeast has been well studied (Figure 2). A PAS is a transient structure formed on yeast vacuoles [41]. An early PAS is an Atg1 complex composed of Atg1, Atg13, Atg17, Atg29, and Atg31, which gradually matures by recruiting downstream ATG proteins and vesicles, forming the initiation site of autophagy [3,40,42]. The PAS-forming protein complex contains a rich IDR domain [3], and the pattern of PAS formation is highly consistent with the molecular aggregates formed by LLPS. Yuko Fujioka’s team showed that a PAS is a molecular condensate, and its formation is regulated by LLPS [40]. It was found that Atg13 binds to different regions of Atg17 through its Atg17 binding region (17BR) and Atg17 linking region (17LR) and forms early PAS droplets through LLPS to initiate autophagy [3]. The droplets are connected to vacuoles through the Vac8 protein [43], which is consistent with the vacuole localization of the PAS in yeast. This process is regulated by the TOR-mediated phosphorylation of Atg13. Under nutrient-rich conditions, Ser428 and Ser429 of Atg13 are phosphorylated by TOR, and phosphorylated Atg13 impairs its interaction with Atg17 and inhibits the LLPS-mediated formation of PASs [44]. In addition, the LLPS of Atg13 and Atg17-Atg29-Atg31 increases Atg1 kinase activity, leading to autophagy activation, and activated Atg1 further inhibits PAS formation by phosphorylating Atg13 [39,40]. This process can be inhibited by the phosphatase Ptc2- and Ptc3-mediated dephosphorylation of Atg1 and Atg13 [45].

It has been found that FIP200 also has LLPS in mammalian cells, and when the K276 site of its IDR region is acetylated by the acetyltransferase CREBBP, the protein stability is increased to strengthen its LLPS characteristics further and promotes the formation of autophagosomes [46]. In addition, LLPS affects TOR activity and thus affects the formation of autophagy precursors in yeast. This phenomenon also occurs in mammalian cells and is regulated by the kinase DYRPK3 [47]. In summary, the important role of LLPS in the initiation of yeast autophagy may provide new insight into the mechanism underlying autophagy initiation. However, there may still be some questions about the role of LLPS in autophagy initiation in mammalian cells. For example, does the mammalian ULK1 complex have a similar LLPS phenomenon to the Atg1 complex in yeast? Do other autophagy core proteins in mammalian cells have identical behavior to that of yeast? Is the LLPS of the Atg1 complex regulated by post-translational modifications? These questions need to be answered by further investigation.

## 3. The Role of Endoplasmic Reticulum (ER) in the Initiation of Autophagy

The formation of autophagosomes is the core event of autophagy initiation, but the membrane source of autophagosomes in multicellular organisms has been debated. According to previous studies, autophagosome has several membrane sources, including the plasma membrane [48], mitochondria [49], endosomes, Golgi, and ER [50,51].

It has been proposed that the ER is the initiation site of autophagosomes (Figure 3) [52,53,54]. Multiple studies observed the formation of isolation membranes (IMs), the membrane structure of pre-autophagosomes, occurring at the ER [54,55,56,57]. The omega-shaped structures associated with the ER, known as omegasomes, are thought to serve as platforms for phagophore assembly by recruiting essential proteins such as DFCP1/ZFYVE1 and facilitating lipid transfer to expand the phagophore [58]. In addition, the COP-II vesicles at the ER function as a membrane source for the formation of autophagosomes [55]. The autophagic initiation complex consisting of FIP200/ATG13/ULK1 directly interacts with the ER protein VAPA/B and forms the initiation site of autophagy in the ER (Figure 3) [59]. The ER-localized transmembrane proteins Atlastin 2 and 3 (ATL2/3) are also involved in this process by helping ULK1 and ATG101 to recruit FIP200 and ATG13 to form the autophagy initiation complex [60]. The phosphatidylinositol 3-kinase catalytic subunit 3 (PI(3)KC3) subsequently produces phosphatidylinositol 3-phosphate (PI(3)P) to recruit autophagic proteins to form omegasomes for autophagosome maturing [58]. Although many studies indicate that the ER plays an important role in autophagy initiation, the signal that determines the initiation of autophagosomes in the ER has not been identified. One report observed that stimulation of autophagy triggers a transient Ca2+ enrichment on the outer surface of the ER membrane, causing FIP200-mediated LLPS to form an autophagy initiation complex [61]. This process is controlled by the ER transmembrane autophagy protein EPG-4/EI24. When EPG-4/EI24 is missing, the autophagosome formation is impaired upon continuous Ca2+ oscillation on the outer surface of the ER [61].

## 4. The Role of ATG9 Vesicles in the Initiation of Autophagy

ATG9 is the only transmembrane ATG protein identified so far. It exists as a trimer in the membrane vesicles [62]. ATG9 has been identified as a phospholipid scramblase that can redistribute phospholipids in both layers of the membranes [63,64]. ATG9 vesicles are initially formed in the ER and transported to the Golgi for maturation [65,66]. The matured ATG9 vesicles are derived from the trans-Golgi network (TGN) and transported to the target sites for autophagy initiation [65,66].

It has been demonstrated that ATG9 vesicles are required for autophagy [67]. The function of ATG9 vesicles in autophagy has been studied extensively. It was proposed that ATG9 vesicles serve as the membrane source for the expansion of phagophores during autophagy initiation [65].

Recent breakthrough studies using the in vitro reconstitution method with yeast autophagic proteins discovered that Atg9 vesicles function as the seeds of the phagophores during the initiation of autophagy [68] (Figure 4). In this model, Atg9 vesicles may interact with the Atg1 (ULK1) complex and other autophagic proteins to obtain phospholipids from the ER membrane or other membrane sources to expand the vesicles. Atg2, a lipid transfer protein, is recruited between the ER membrane and Atg9 vesicles [69,70,71]. At the same time, the PI3KC3 complex (PI3KC3-C1) may be recruited to ATG9 vesicles and subsequently the ATG12-ATG5-ATG16L complex for Atg8 lipidation [24]. The expansion of ATG9 vesicles into the phagophores is processed by transferring the phospholipids from the ER membrane to the Atg9 vesicle through the Atg2 lipid transfer channel [69,70,71]. The phospholipids transferred from the ER membrane are accepted and integrated into the ATG9 vesicle membrane by Atg9 with its phospholipid scramblase activity [63,64]. At the same time, Atg8 may be lipidated with phosphatidylethanolamine (PE) in the membrane of ATG9 vesicles along with the expansion. In this way, ATG9 vesicles function as the seeds of autophagosomes by the formation of the isolation membrane or the phagophore, which is the initiation structure of autophagosomes (Figure 4). Although this in vitro model is established with yeast autophagic proteins, a similar autophagic initiation mechanism is likely to exist in mammalian cells.

ATG9 vesicles may also function as the seeds of autophagosomes during the initiation of selective autophagy. ATG9 is capable of interacting with both autophagy receptors and the autophagic initiation complex, such as SQSTM1 and the ULK1 complex [24,72]. Thus, ATG9 vesicles target both the selective autophagic cargos and the source membranes and initiate the expansion of the vesicles to form the PAS [24].

Many questions remain unanswered regarding the role of ATG9 in the initiation of autophagy. For example, how are ATG9 vesicles transported to the initiation site in response to the autophagy activation signaling? How are the autophagy initiation complexes, including the PI3KC3 complex, the LC3 lipidation complex, and the ATG2-ATG9 complex, assembled and organized at the vesicles in response to autophagy activation? Does the ATG9 vesicle also function as a membrane source in the expansion of the PAS in the initiation of autophagy? These interesting questions need further investigation to be answered to further our understanding of the mechanisms underlying the initiation of autophagy.

## 5. The Cellular Signaling Pathways That Regulate Autophagy

Autophagy functions to mitigate various cellular stresses and protect cells from stress-caused damage. Thus, the signaling pathways involved in the initiation and activation of autophagy are largely related to cellular stresses. The mTOR-related signaling pathway is the first one found to directly regulate the activation of autophagy in response to nutritional stress or other metabolic stress conditions. In recent years multiple non-mTOR signaling pathways have been identified to be involved in the activation of autophagy. The cellular signaling pathways that regulate the initiation and activation of autophagy are summarized in Figure 5.

### 5.1. The mTOR-Dependent Signaling Pathways in the Regulation of Autophagy

The mTOR participates in multiple metabolic signaling pathways [73]. mTOR signaling is a major negative regulator of autophagy and plays an important role in the regulation of autophagy activation [74].

#### 5.1.1. The AKT-TSC-RHEB—mTOR Signaling Pathway

RHEB is a small GTPase that is capable of directly regulating mTOR kinase activity [75]. The active RHEB directly binds to the kinase domain of mTOR and activates its kinase activity, thus playing an inhibitory role in the regulation of autophagy [76]. The RHEB is inactivated by the TSC1/2 complex, which is the GTPase-activating protein (GAP) of RHEB [75]. However, the RHEB GAP activity of the TSC1/2 complex is negatively regulated by the AKT kinase-mediated phosphorylation of TSC2 [77,78,79,80,81,82]. Thus, the activation effect of RHEB on mTOR kinase activity, which leads to the inhibitory effect on autophagy, is positively regulated by AKT and negatively regulated by the TSC1/2 complex [77,78,79,80,81,82]. Recent studies have shown that RHEB-mediated mTORC1 activation is regulated by ubiquitination [83].

#### 5.1.2. The AMPK-TSC1/2-m TOR Signaling Pathway

AMP-activated protein kinase (AMPK) is a conserved serine/threonine protein kinase and a key metabolic sensor in cells [84]. When the intracellular ATP/AMP ratio decreases, increased AMP activates AMPK activity by binding to the AMPK regulatory subunit γ [85,86,87], which causes allosteric change and the phosphorylation of the Thr172 site [87,88]. Activated AMPK promotes GAP activity of the TSC1/2 complex by directly phosphorylating TSC2 at T1227 and S1345, thus inactivating RHEB and mTOR and activating autophagy [89,90,91]. Intracellular calcium also participates in the AMPK-induced autophagy. An increase in the intracellular Ca2+ concentration activates CaMKK2 kinase [92]. Activated CaMKK2 activates AMPK by the phosphorylation of AMPK at the Thr172 site and triggers autophagy in response to changes in the calcium flux [93,94].

#### 5.1.3. The Ras/Raf-MEK-ERK Signaling Pathway

Recent studies have shown that the Ras/Raf-MEK-ERK signaling pathway regulates autophagy [95]. It has been found that the inhibition of the KRAS-RAF-MEK-ERK signaling induces autophagy, as the inhibition of MEK1/2 leads to the activation of the LKB1-AMPK-ULK1 axis [96]. In addition, the ERK1/2-RSK pathway mediates the phosphorylation of the TSC1/2 complex and inhibits its GAP activity, leading to the activation of RHEB and mTORC1 [97,98]. However, some studies have shown contradictory results, in which the activation of the Ras/Raf-MEK-ERK pathway induces autophagy [99,100], suggesting that the regulation of autophagy by the Ras/Raf-MEK-ERK pathway is complicated and the mechanism needs to be further investigated.

#### 5.1.4. The RAG GTPase Signaling Pathway

Recent studies have found that amino acid deprivation activates autophagy by the inhibition independent of the theTSC1/2 complex; instead, it is dependent on RAG small GTPases [101,102,103,104,105]. There are four RAG GTPases in mammalian cells: RAGA, RAGB, RAGC, and RAGD. Either RAGA or RAGB forms a heterodimer with either RAGC or RAGD. The RAG heterodimers are associated with lysosomal membrane via a lysosomal protein complex composed of p18 (also known as LAMTOR1), p14 (LAMTOR2), MP1 (LAMTOR3), C7orf59 (LAMTOR4), and HBXIP (LAMTOR5) [106]. Amino acid enrichment triggers the active form of RAG heterodimers (RAGA/B-GTP with RAG-C/D-GDP), which recruits mTORC1 to lysosomes [107,108]. At lysosomes, mTORC1 is further activated by RHEB [104,109,110]. Conversely, amino acid deprivation leads to the inactivation of RAG heterodimers (RAGA/B-GDP with RAGC/D-GTP), which dissociates mTORC1 from lysosomes and inactivates mTORC1 (Figure 5).

#### 5.1.5. The GSK3β-TSC Signaling

It has been found that GSK3β, an important negative regulator of the Wnt signaling pathway, phosphorylates and activates the TSC1/2 complex, resulting in the inhibition of mTOR kinase activity and the activation of autophagy [111,112]. Early studies observed that GSK3b was phosphorylated by Akt [113], suggesting that the role of GSK3β in the activation of autophagy may be regulated by Akt-mediated phosphorylation. Indeed, the phosphorylation of GSK3β by Akt inhibits autophagy, while the dephosphorylation of GSK3β enhances autophagy [114,115]. Thus, the Wnt signaling pathway may inhibit autophagy through the phosphorylation and inhibition of GSK3β.

### 5.2. The mTOR-Independent Signaling Pathways

#### 5.2.1. The mTOR-Independent AMPK Signaling

AMPK phosphorylates multiple sites of ULK1, such as Ser467 and Thr574, to activate ULK1, thereby activating autophagy [116,117]. In addition, AMPK activates autophagy by regulating the PI3KC3 complex. During glucose starvation, AMPK binds to VPS34 and ATG14L, phosphorylates BECN1, and stimulates autophagy activity [118]. Interestingly, AMPK determines the differential function of the PI3KC3 complex in autophagy [119,120]. AMPK phosphorylates the T163/S165 sites of VPS34 to inhibit the non-autophagic function of the PI3KC3 complex and phosphorylates the S91/S94 sites of BECN1 to activate the pro-autophagic function of the PI3KC3 complex and induce autophagy [120]. This illustrates the role of AMPK, which switches between the non-autophagic or pro-autophagic functions of the PI3KC3 complex. Intracellular calcium participates in AMPK-induced autophagy. An increase in the intracellular Ca2+ concentration leads to the activation of CaMKK2 kinase [92]. Activated CaMKK2 activates AMPK by the phosphorylation of AMPK at the Thr172 site and then triggers autophagy in response to changes in the calcium flux [93,94].

#### 5.2.2. The Bcl-2 Signaling in Autophagy

The Bcl-2 protein family plays important roles in the regulation of apoptosis [121]. It has been shown that anti-apoptotic Bcl-2 proteins, such as Bcl-2, Bcl-XL, Bcl-w, and Mcl-1 inhibit autophagy by interacting with the autophagic protein BECN1 [122,123,124,125,126]. The anti-apoptotic Bcl-2 proteins bind to the BH3 domain of BECN1, thus inhibiting the PI3K activity of the PI3KC3 complex [127]. The BH3-only pro-apoptotic proteins, such as BNIP3, Bad, Bik, BimEL, Noxa, and Puma, induce autophagy by competing with the binding of BECN1 to the anti-apoptotic Bcl-2 proteins and releasing the inhibition of BCEN1by the anti-apoptotic Bcl-2 proteins [126,128,129,130,131]. In addition, it has been proved that activated c-jun terminal protein kinase 1 (JNK1) induces autophagy by phosphorylating Bcl-2 and dissociating Bcl-2 from the BECN1 complex [132].

### 5.3. The Death-Associated Protein Kinase (DAPK) Signaling Pathway

Recently, it has been found that death-associated protein kinase 3 (DAPK3) directly phosphorylates ULK1 at Ser556, and promotes the formation of the ULK1 autophagy initiation complex, thus activating autophagy [133]. In addition, DAPK phosphorylates BECN1 at Thr119 in the BH3 domain, leading to the dissociation of BECN1 from Bcl-XL and the activation of autophagy [134].

### 5.4. The O-GlcNAcylation Pathway in Autophagy Initiation

O-GlcNAc is a protein post-translational modification that uses uridine diphosphate N-acetylglucosamine (UDP-GlcNAc) as a donor molecule to modify the hydroxyl group of serine or threonine of the target proteins [135]. Multiple studies have shown that O-GlcNAcylation activates autophagy [136,137]. O-GlcNAcylation of the autophagy regulatory proteins regulates their activity, stability, and subcellular localization, thus regulating autophagy [138]. Studies have shown that the GlcNAc modification of AKT Thr305 and Ser312 sites antagonizes Ser308 phosphorylation [139], and mouse experiments have shown that OGT (O-GlcNAc transferase) content is negatively correlated with Akt expression [140]. This indicates that O-GlcNAc modification is likely to affect mTOR activation by regulating Akt activity and expression, thereby regulating autophagy initiation. In addition, AMPK, another key protein in the autophagy activation pathway, is also modified by O-GlcNAcylation, which inhibits its kinase activity [141]. Furthermore, the ULK1/2 O-GlcNAcylation level is positively correlated with autophagic activity [76]. These studies indicate that O-GlcNAcylation is a novel regulatory means for autophagy.

## 6. Conclusions

The initiation and activation of autophagy are the most important steps for autophagic processes. LLPS is the newly identified biophysical process for the assembly of a pre-autophagic complex, and the ER is a major cellular location for the initiation of autophagy and the formation of autophagosomes. Atg9 vesicles have been identified as the seeds for autophagosomal biogenesis and are the crucial component for the initiation of autophagy. The activation of autophagy is regulated by multiple signaling pathways that are in response to cellular stress conditions. As shown in Figure 5, these signaling pathways are centralized by the ULK1 complex, the mTOR complex, the AMPK complex, and the PI3KC3 complex. Interactions with these functional complexes constitute the regulatory networks for the initiation and activation of autophagy.

Currently, many questions remain unanswered regarding the regulation of the initiation and activation of autophagy. For example, how does the autophagy initiation complex assemble temporally and spatially in response to the autophagic activation signal? How does the autophagy initiation complex determine or recognize the initiation site? How do the multiple autophagy regulatory signaling pathways coordinate in response to nutritional or oxidative stress in cells? Furthermore, as the function of Atg8/LC3 is expanded from findings of ATG8ylation, other cellular processes such as endosomal trafficking and vesicle secretion may participate in the regulation of or coordination with autophagy initiation and activation. Further investigation of these questions will greatly improve our understanding of molecular mechanisms underlying autophagy initiation and activation.

## Figures and Tables

**Figure 1 biomolecules-14-01517-f001:**
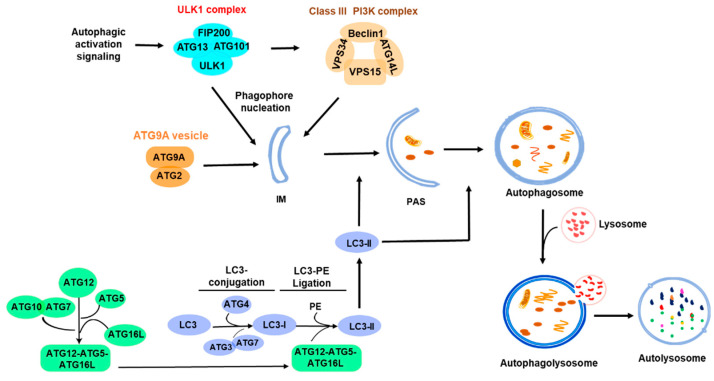
Overview of autophagy process. The ULK1 complex is activated under various autophagy signals to initiate autophagy, and the nucleation process is completed with the participation of the PI3K complex and the ATG9A system. Furthermore, the extension of the autophagic membrane is achieved through the two ubiquitin-like ATG conjugation and ligation processes: the ATG3-ATG7-mediated LC3 conjugation and the ATG12-ATG5-ATG16L-mediated LC3 ligation with phosphatidylethanolamine (PE). The PAS develops into an autophagosome that eventually fuses with a lysosome for degradation. The shaded area indicates the initiation part of the autophagic processes. IM, isolation membrane; PAS, pre-autophagosomal structure; PE, phosphatidylethanolamine.

**Figure 2 biomolecules-14-01517-f002:**
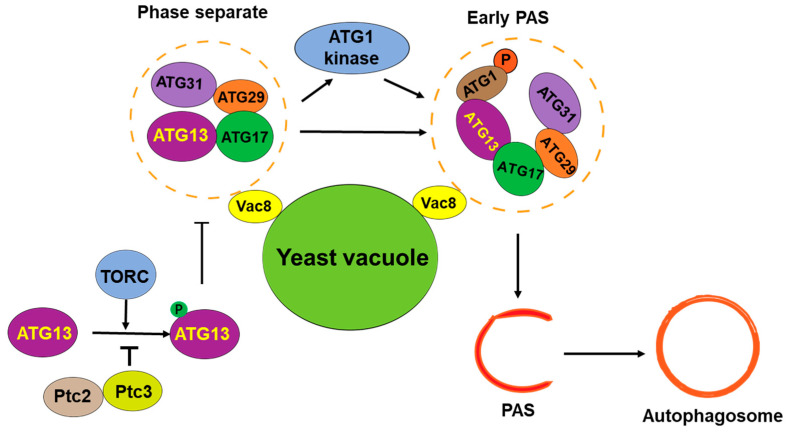
LLPS in assembly of PAS in yeast cells. The protein complex of autophagy initiation is assembled through phase separation and organized into early PAS. At the same time, TORC, an important TOR protein complex, regulates the phase separation of the autophagy initiation complex.

**Figure 3 biomolecules-14-01517-f003:**
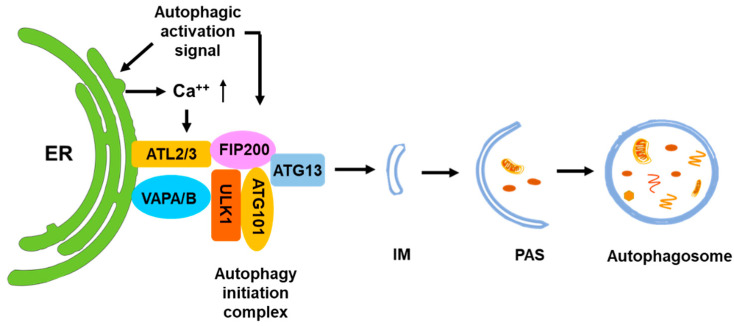
ER is the major initiation site for autophagy. The ULK1 complex is assembled at ER in response to the autophagic activation signal and binds to the ER membrane through interaction with ATL2/3 and VAPA/B. The ER-bound ULK1 complex recruits ATG9 vesicles and other autophagic initiation complexes for initiation of autophagy. Calcium signaling of ER facilitates the assembly of the ULK1 complex at ER.

**Figure 4 biomolecules-14-01517-f004:**
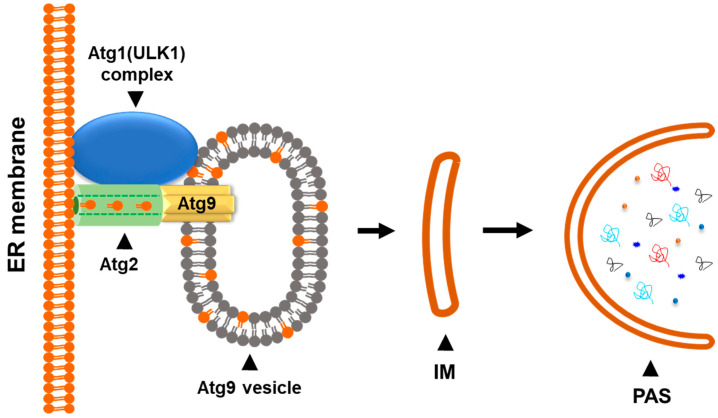
The Atg9 vesicle functions as a seed for autophagosomal biogenesis. The Atg9 vesicle may interact with the Atg1 (ULK1) complex and other autophagic proteins in response to the autophagic initiation signal. The phospholipid transfer protein Atg2 connects the phospholipid source membrane (ER membrane) and the Atg9 vesicle. Expansion of Atg9 vesicle into the isolation membrane (IM) is processed by the Atg2-mediated phospholipid transferring and the Atg9-mediated phospholipid scrambling.

**Figure 5 biomolecules-14-01517-f005:**
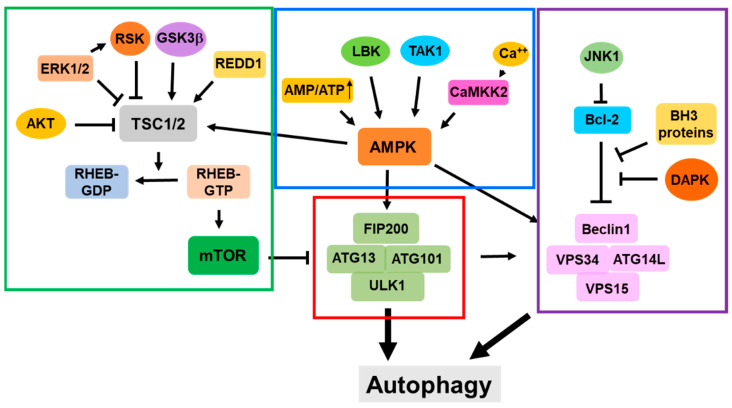
Cellular signaling pathways for regulation of initiation and activation of autophagy. The red line circled portion is the ULK1 complex; the green line circled portion shows the mTOR complex-related signaling pathways; the blue line circled portion shows the AMPK complex-related signaling pathways; and the purple line circled portion shows the PI3KC3 complex-related signaling pathways.

## Data Availability

Not applicable.

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
