# Peer review of "Molecular Mechanisms Underlying Initiation and Activation of Autophagy"

_biomolecules, 2024, doi:10.3390/biom14121517_

Round 1

Reviewer 1 Report (Previous Reviewer 3)

Comments and Suggestions for Authors

The authors ignored a large portion of my comments and did not address them at all.

Author Response

Reviewer 2 Report (Previous Reviewer 2)

Comments and Suggestions for Authors

the present form of manuscript is benefitial for readers

Author Response

Comment 1: the present form of manuscript is benefitial for readers.

Response 1: Many thanks for your positive comment.  We are very encouraged by your comments and will keep our interest in the research progress in autophagy.

Reviewer 3 Report (Previous Reviewer 1)

Comments and Suggestions for Authors

Wei et al. review recent advancements in understanding autophagy initiation and activation, a crucial catabolic process for cellular homeostasis. The authors highlight groundbreaking discoveries, including the role of liquid-liquid phase separation (LLPS) in initiating autophagy and the identification of specific organelles like the endoplasmic reticulum and Golgi apparatus as key initiation sites. The review also delves into recent insights on critical signaling pathways regulating autophagy, such as the ULK1, mTOR, AMPK, and PI3KC3 complexes, revealing new regulatory mechanisms and interactions.

I would like to thank the authors for incorporating my previous suggestions. The addition of the new figures brings real added value to their review, especially as they are both clear and easy to understand. The improvement in layout, as well as the quality and formatting of the references, is also appreciated. However, I noticed one last typo on line 240, where the word pathway is written twice. No mention is made of ATG8ylation, which does not detract from the quality of the review but could be considered for future publications, especially as this field is emerging. Including this topic in the future could provide an even more comprehensive perspective on the subject.

Author Response

Comment 1: Wei et al. review recent advancements in understanding autophagy initiation and activation, a crucial catabolic process for cellular homeostasis. The authors highlight groundbreaking discoveries, including the role of liquid-liquid phase separation (LLPS) in initiating autophagy and the identification of specific organelles like the endoplasmic reticulum and Golgi apparatus as key initiation sites. The review also delves into recent insights on critical signaling pathways regulating autophagy, such as the ULK1, mTOR, AMPK, and PI3KC3 complexes, revealing new regulatory mechanisms and interactions.

Response 1: Thank you very much for your valuable advice. Your comments are very helpful for us to improve quality of the manuscript.

Comment 2: I would like to thank the authors for incorporating my previous suggestions. The addition of the new figures brings real added value to their review, especially as they are both clear and easy to understand. The improvement in layout, as well as the quality and formatting of the references, is also appreciated. However, I noticed one last typo on line 240, where the word pathway is written twice.    

Response 2: Thanks for the critiques. We have corrected the error in the revised manuscript.

Comment 3: No mention is made of ATG8ylation, which does not detract from the quality of the review but could be considered for future publications, especially as this field is emerging. Including this topic in the future could provide an even more comprehensive perspective on the subject.

Response 3:  Many thanks for your suggestion. ATG8ylation (LC3ylation in mammalian cells) is a new concept for the ATG8 family protein-mediated membrane and protein modification. It emerges as a growing point for exploring new cellular function of the ATG8 family proteins.  Based on your suggestion, we have added a portion to briefly introduce ATG8ylation in the section of Overview of autophagy (please see the red portion in Overview of autophagy).

Round 2

Reviewer 1 Report (Previous Reviewer 3)

Comments and Suggestions for Authors

The authors addressed my concerns.

This manuscript is a resubmission of an earlier submission. The following is a list of the peer review reports and author responses from that submission.

Round 1

Reviewer 1 Report

Comments and Suggestions for Authors

Wei et al. review recent advancements in understanding autophagy initiation and activation. The authors highlight recent discoveries, including the role of liquid-liquid phase separation in initiating autophagy and the identification of specific organelles like the endoplasmic reticulum as key initiation site. The review also delves into recent insights on critical signaling pathways regulating autophagy, such as the ULK1, mTOR and AMPK.

The original figures that punctuate the text are particularly appreciable, as they effectively illustrate the complex concepts discussed. However, the absence of Figure 2 is to be noted and should be corrected to ensure the coherence of the whole.A strong point of this review is in the presentation of open questions at the end of each section. This structure encourages critical reflection and potentially stimulates new research tracks.

In summary, this review constitutes a precious resource for researchers in the field, offering a complete and up-to-date overview of autophagy initiation and activation mechanisms, while underlining the questions that rest to be explored.

This review could also be improved by addressing the following concerns:

Overall presentation:

-        Please review the document for unnecessary spaces that may visually clutter the text and impede readability.

-        There appears to be a dual numbering system for references, which reduces readability. Please standardize the referencing system in the reference part.

-        Consider including an initial figure to illustrate the canonical autophagy flux (with and without selective autophagy) and the major associated complexes presented in the text (for both mammalian and yeast systems). This visual aid would enhance reader understanding of the processes described.

Specific comments:

-        Lines 24-31: Consider refining your referencing strategy for improved clarity and comprehensiveness. Please:

o   Include a more general reference encompassing all three types of autophagy on line 26.

o   Line 26, it seems that a verb is lacking in the sentence

o   Provide a specific review reference for each autophagy type as they are mentioned in lines 26-28. (Currently, microautophagy lacks a dedicated reference.)

o   References 6-7 could be replaced with more comprehensive reviews on autophagy, rather than focusing only on autophagosome maturation or biogenesis.

-        Line 38: Consider replacing the current general autophagy references with sources more specifically focused on selective autophagy. I suggest including a review by Terje Johansen, who has published extensively on this topic.

-        Line 40: For consistency and to avoid potential confusion, I recommend adhering to a single nomenclature system throughout the manuscript. In this case, you should really be precise between mammalian and yeast, especially since you are using both systems in your review. This part of the sentence could also benefit from a well-chosen reference.

-        Line 42: Please note that the conventional naming for mammalian ATG16 is ATG16L1.

-        Lines 70-71: Please cite reference 34 before reference 35 in the text to maintain chronological order.

-        Line 69: The sentence seems grammatically incorrect. Please revise for clarity and proper structure.

-        Reference 35: This reference does not adequately describe the ATG1/PAS complex and focuses more on ATG9. Consider replacing it with a more relevant source that specifically addresses the ATG1/PAS complex.

-        Line 73: A reference is missing here. Please add an appropriate citation to support the statement.

-        Line 84: There is a minor typographical error (a missing space). Please correct this for improved readability.

-        Line 88: it seems there is a missing word. Please check and complete to ensure clarity.

-        Part 3: It would be beneficial to introduce the concept of omegasomes before using it in the text. It's important to specify whether this section pertains to mammals or yeast, as the current context is confusing.

-        Figure 2 is missing, making it impossible to evaluate. Please include it to allow for a complete analysis. More over, Figure 2 Is never mentioned in the text.

-        Line 189: The word "complex" or "signalling" is likely absent.

-        Line 199: There is a noticeable jump in the reference numbering, from 66 to 73. Please revise the order of references throughout the document. Ensure that all references are introduced sequentially and logically.

-        Line 221: The word "signaling" is repeated twice consecutively.

Reviewer 2 Report

Comments and Suggestions for Authors

The contribution is weak over the state of the art. The methodology lacks a proper analysis.

Comments on the Quality of English Language

The language is fluent

Reviewer 3 Report

Comments and Suggestions for Authors

The review aims to describe autophagy initiation and activation, important stages of the autophagy pathway. Unfortunately, the presented manuscript is poorly executed.  Words are missing in sentences (e.g. line 26, line 88). The entire Figure 2 is missing in the manuscript! Information in the manuscript is inaccurate in many places. For example, the authors use incorrect definition of the well-known abbreviation “IDR”.  The ATG5~ATG12 complex is incorrectly depicted in Figure 3 that it functions on its own. In reality, this complex binds ATG16L1 to operate as an E3-like enzyme. The term ATG7-ATG3 conjugation complex in Figure 3 is misleading and inaccurate because these two proteins function as E1-like and E2-like enzymes in the ATG8/LC3/GABARAP conjugation system. LC3 conjugation system is incorrectly listed as a part of the autophagy initiation. Short paragraphs in the 5th Section (lines 181-299) are very condensed fragmented pieces of information written with little fluency, almost like notes, and are difficult to understand. Questions at the end of paragraphs are not original or thoughtful. They are obvious, long-standing questions in the autophagy field. The review is superficial, and I do not consider it a meaningful contribution on the topic of autophagy.

Comments on the Quality of English Language

See above, no further comments.